# Effect of Potato Glycoside Alkaloids on Energy Metabolism of *Fusarium solani*

**DOI:** 10.3390/jof9070777

**Published:** 2023-07-24

**Authors:** Chongqing Zhang, Dedong Ding, Bin Wang, Yupeng Wang, Nan Li, Ruiyun Li, Yuke Yan, Jing He

**Affiliations:** 1College of Forestry, Gansu Agricultural University, Lanzhou 730070, China; zhangchongqing2022@163.com (C.Z.); 15214045659@163.com (D.D.); wangbin_1519@163.com (B.W.); 18294995796@163.com (Y.W.); lilinannan0709@163.com (N.L.); lruiyun@163.com (R.L.); xlsyyk@163.com (Y.Y.); 2Wolfberry Harmless Cultivation Engineering Research Center of Gansu Province, Lanzhou 730070, China

**Keywords:** potato glycoside alkaloids, *Fusarium solani*, energy metabolism, mitochondria

## Abstract

*Fusarium solani* is one of the primary pathogens causing root rot of wolfberry. The aims of this study were to investigate the inhibitory effect of potato glycoside alkaloids (PGA) on *F. solani* and its energy metabolism. In this study, the effects of PGA treatment on the growth and development of *F. solani* were investigated and the changes in the glycolytic pathway (EMP), ATPase activity, mitochondrial complex activity, mitochondrial structure, and energy charge level were analyzed to elucidate the possible antifungal mechanism of PGA on *F. solani*. The results showed that PGA treatment inhibited the colony growth, biomass, and spore germination of *F. solani*. PGA treatment reduced the glucose content and Hexokinase (HK) activity of *F. solani*, but increased the activity of Fructose-6-Phosphate Kinase (PFK) and Pyruvate Kinase (PK) and promoted the accumulation of pyruvic acid. In addition, PGA treatment inhibited the activities of H^+^-ATPase, Ca^2+^-ATPase, and mitochondrial complex IV, increased the mitochondrial inner membrane Ca^2+^ content and mitochondrial membrane permeability transition pore, and decreased the contents of ATP, ADP, and AMP as well as the energy charge. These results indicate that PGA treatment inhibits the growth and development of *F. solani*, activates the glycolysis pathway, inhibits ATPase activity and mitochondrial complex activity, and destroys the structure and function of mitochondrial membrane, resulting in a lower energy charge level.

## 1. Introduction

*Fusarium solani* is a filamentous pathogenic fungus widely distributed in the world. The fungus is the main pathogen of plant soil-borne diseases and causes a variety of plant rots, such as root rot, stem rot, ear rot, and stem base rot [1,2]. Among these, root rot is one of the main diseases of wolfberry in Gansu Province, China. The disease is not easily detected early on, and is difficult to cure when the root tissue decays. It is a devastating disease that seriously threatens the development of the wolfberry industry. Currently, the prevention and control of wolfberry root rot are mainly based on chemical fungicides. However, due to the difficulty of prevention and control of this disease and the threat posed by chemical fungicides to human health and soil ecology, it is of significant interest to develop safe and environment friendly methods to replace or reduce chemical fungicides [3].

Potato glycoside alkaloids (PGA) are nitrogen-containing and odorous steroidal glycoalkaloids formed by combining 1–4 monosaccharides through 3-0-glycosidic bonds. PGA is a ligand of solanidine with glucose, lactose, and rhamnose, respectively. The crystal is white light-emitting, needle-like, easily soluble in methanol and hot ethanol, slightly soluble in cold ethanol, and almost insoluble in water, ether, and benzene, with a melting point of 248 °C and decomposition at about 280–285 °C; it can be hydrolyzed by acid and decomposes into solanidine and sugar. The structure is shown in Figure 1 [4]. It is mainly found in potato buds and skins. Because of its comprehensive source of material, low cost, and no harm to the environmental, it has gradually become a research hotspot [5,6]. PGA has a wide range of biological activities, and over 90% of its content is α-solanine and α-chaconine [7]. Previous studies have shown that PGA can inhibit the growth of *Pectobacterium carotovorum* as well as weaken the vitality of *Pectobacterium bacterial* and reduce its pathogenicity [8,9]. PGA inhibited the mycelium growth of *Pyrenophora teres*, *Alternaria alternata*, and *Phytophthora infestans* and the spore germination of *Curvularia trifolii* [10,11,12], and was able to kill part of the mycelium of *Fusarium sulphureum* and inhibit its growth [13].

Mitochondria are the prominent organelles of cell energy metabolism and are an essential regulator of cell survival, signal transduction, differentiation, and cell cycle progression. They mainly produce ATP through intracellular glycolysis (EMP), the tricarboxylic acid cycle (TCA), and oxidative phosphorylation [14,15]. Structural damage or dysfunction of the mitochondria can significantly change the energy level of pathogenic fungi, affecting cell adaptability and leading to cell death [16]. Our previous studies have found that PGA treatment can inhibit the growth and development of *F. solani* by destroying its cell structure, interfering with substance metabolism, and inhibiting respiration and reactive oxygen species metabolism [17,18]. However, the effect of PGA on mitochondrial energy metabolism in *F. solani* is unclear.

Therefore, in this study we observed the effects of PGA on the biomass and spore germination of *F. solani*, determined the changes in key enzyme activities and product content in the glycolytic pathway, and analyzed the effects of PGA treatment on mitochondrial complex I-IV activity and mitochondrial structure and energy charge level in order to reveal the possible antifungal mechanism of PGA against *F. solani* from the perspective of energy metabolism.

## 2. Materials and Methods

### 2.1. Materials

*Fusarium solani* (Mart.) App. et Wollenwwas belongs to the Tuberculariaceae. It was isolated from diseased wolfberry plants affected by root rot. *F. solani* was identified by morphology and EF-1α gene sequence analysis and comparative identification, and was preserved in the Forest Protection Laboratory of Gansu Agricultural University.

The potatoes (*Solanum tuberosum* L. cv. Longshu No. 9) were harvested from Tongwei County of Dingxi City, Gansu Province in China in October 2022. Fresh potatoes were washed and exposed to sunlight in order to accelerate the process of greening and germination. The green potato peels and buds were dried in a vacuum blast drying oven (80 °C, 12 h), ground using a grinder, and passed through an 80-mesh sieve. The obtained experimental samples were extracted by the acetic acid–ammonia precipitation method [17]; 100 g of the potato sample was mixed with 400 mL of 5% acetic acid, stirred for 60 min, and filtered. The residue was extracted twice with 200 mL of 5% acetic acid, the filtrate was combined, and its pH was adjusted to 11 with ammonia. After three extractions with 200 mL of water-saturated n-butanol, the extracts were combined and dried on a rotary evaporator and the residue was mixed with 20 mL of methanol to obtain the total glycoalkaloids extract.

Referring to the results of Li et al., PGA was identified by high-performance Liquid chromatography (HPLC) [13,19]. The PGA was then dissolved in methanol and stored at 4 °C for later use.

### 2.2. Preparation of F. solani Spore Suspensions

A spore suspension of *F. solani* was prepared according to the method of Lachhab et al. [20]. After seven days of culture, 10 mL of sterile water was added to the *F. solani* plate and two drops of Tween-80 were dropped. The spores were shaken, scraped, transferred to a conical flask, and filtered through a four-layer gauze. The concentration was diluted to 1 × 10^6^ spores·mL^−1^ using a hemocytometer and stored at 4 °C.

### 2.3. Antifungal Effect of PGA

The antifungal diameter was determined according to the method of Yan et al. [21].

The mycelium plug (5 mm) of *F. solani* was placed in the center of potato dextrose agar (PDA) and four Oxford cups were placed 2 cm away from the mycelium plug. The PGA was added dropwise (20 μL 88.10 mg·mL^−1^) to each Oxford cup; the same amount of sterile distilled water was used as a control and incubated in a 25 °C incubator for 7 days. The diameter was measured by the crossover method.

### 2.4. Determination of Biomass of F. solani

The biomass was measured according to the method of Li et al. [22]. Previously, we found that the EC_50_ value of PGA inhibiting *F. solani* was 4.432 7 mg·mL^−1^ [18]. The 0.5 g of fresh mycelium was inoculated in PDB (150 mL conical flask containing 30 mL of PDB medium) containing (4.432 7 mg·mL^−1^ PGA) medium for 36 h. At the same time, sterile water was used as the control. The mycelium was collected at 0, 3, 6, 9, 12, 24, and 36 h, centrifuged at 4 °C, 10,000× *g* for 10 min, washed three times with sterile water, dried to constant weight at 60 °C, and weighed.

### 2.5. Determination of Spore Germination Rate of F. solani

The spore germination rate was determined according to the method of Li et al. [23]. PGA was mixed with prepared spore suspension of *F. solani* according to mass concentration EC_50_, 1/2 EC_50_, 1/4 EC_50_, 1/8 EC_50_ and 1/16 EC_50_, and sterile water was set as control. After mixing the above solution, the appropriate amount of mixed solution was suspended on the concave glass plate with a pipette gun and moisturized at 25 °C. Spore germination was counted at 0, 3, 6, 9, and 12 h, respectively. Each treatment was repeated three times and the spore germination rate was calculated for each microscopic examination of 150–200 spores. The spore germination rate was calculated using Formula (1):(1)$$\mathrm{Spore}\;\mathrm{germination}\;\mathrm{rate}\;(\%)\;=\;\frac{\mathrm{Number}\;\mathrm{of}\;\mathrm{germinated}\;\mathrm{spores}}{\mathrm{Total}\;\mathrm{number}\;\mathrm{of}\;\mathrm{spores}\;\mathrm{investigated}}\;\times \;100$$

### 2.6. Determination of Glucose and Pyruvate Contents, Hexokinase (HK), Pyruvate Kinase (PK), and Fructose-6-Phosphate Kinase (PFK) Activities of F. solani

The glucose content was determined according to the method of Yang et al. [24]. The 0.2 g of frozen mycelium was ground in 1 mL of normal saline, centrifuged at 4 °C and 8000× *g* for 10 min, and the supernatant was collected for use. Next, 150 μL of glucose oxidase, 75 μL of titanium oxalate potassium, 175 μL of normal saline, and 100 μL of extract were added to the cuvette. The liquid was immediately placed in an ultraviolet spectrophotometer and the absorbance was measured at a wavelength of 382 nm. The content of glucose in the sample was calculated according to the standard curve, expressed as mg·mL^−1^.

The activities of pyruvate (PA) (YX-C-B204, Sino Best Biological Technology Co., Ltd., Beijing, China), hexokinase (HK) (YX-W-B200, Sino Best Biological Technology Co., Ltd., Beijing, China), pyruvate kinase (PK) (YX-W-B201, Sino Best Biological Technology Co., Ltd., Beijing, China), and fructose-6-phosphate kinase (PFK) (YX-W-B202, Sino Best Biological Technology Co., Ltd., Beijing, China) were determined according to the kit instructions. The content of pyruvic acid was expressed as μg·g^−1^ FW. HK activity was expressed as U·g^−1^ FW, and the unit of enzyme activity was 1 nmol NADPH per minute. The consumption of 1 nmol NADH per minute per gram of tissue PK was considered to make up one enzyme activity unit U, and PK activity was expressed as U·g^−1^ FW. With PFK per gram of tissue catalyzing the conversion of 1 nmol NADH to 1 nmol NAD^+^ per minute in the reaction system as an enzyme activity unit U, PFK activity was expressed as U·g^−1^ FW.

### 2.7. Determination of ATPase Activity of F. solani

The protein content of the mycelium was determined using the Coomassie brilliant blue colorimetric method [25]. The 1 mg·mL^−1^ standard protein solution (10, 20, 30, 40, 50, 60 μL) was added to a test tube supplemented with PBS buffer to 100 μL, then 5 mL of Coomassie brilliant blue dye solution was added and shaken. After standing for 5 min, the OD_595_ was determined using a spectrophotometer and the standard curve was drawn. Next, 0.2 g of mycelium was dissolved in 2 mL of PBS, ground in an ice bath, centrifuged at 1000× *g* for 20 min, and 0.1 mL of supernatant was taken for determination.

The H^+^-ATPase and Ca^2+^-ATPase activities of *F. solani* were determined by kits (YX-22463F, Sino Best Biological Technology Co., Ltd., Beijing, China). The operation steps were carried out according to the instructions. The activity unit U was 1 μmol inorganic phosphorus produced by ATP enzyme decomposition of per-mg protein per hour. The activities of H^+^-ATPase and Ca^2+^-ATPase were expressed as U·mg^−1^ protein.

### 2.8. Determination of the Mitochondrial Complex Activity of F. solani

The mitochondria of *F. solani* were extracted by differential centrifugation [26], then 0.1 g of fresh mycelium was added into 1 mL of buffer (Hepes-KOH, 0.6 mol·L^−1^ mannitol, 0.2% bovine serum albumin, 2 mmol·L^−1^ ethylene glycol tetraacetic acid, 2 mmol·L^−1^ dithiothreitol, pH 7.4) and ground in an ice bath. The supernatant was centrifuged at 4 °C, and 12,000× *g* for 10 min, the supernatant was discarded, and the precipitate was the mitochondria.

The prepared spore suspension was added to the PDB medium and cultured at 25 °C, 160 rpm for 48 h in the dark. The mycelium was filtered and washed with sterile water 3–4 times. Accurately weighed 0.5 g hyphae were added to a PDB drug-containing (EC_50_) medium to continue the incubation. At the same time, sterile water was set as the control and the mycelium was taken at 9 h and 36 h for testing. The kit was used to determine the activity of complex I-IV (YX-W-C300, YX-W-C301, YX-W-C302, YX-W-C303, Sino Best Biological Technology Co., Ltd., Beijing, China) and the operation steps were carried out according to the instructions. The consumption of 1 nmol NADH per minute per gram of tissue in the reaction system was used as the enzyme activity unit U and the activity of complex I was expressed as U·g^−1^ FW.

The consumption of 1 nmol 2,6-dichloroindole per minute per gram of tissue in the reaction system was used as the enzyme activity unit U and the activity of complex Ⅱ was expressed as U·g^−1^ FW. The enzyme activity unit U was 1 nmol of reduced cytochrome C per minute catalyzed by complex III per gram of tissue, and the activity of complex III was expressed as U·g^−1^ FW. The catalytic degradation of 1 nmol-reduced cytochrome C per minute per gram of tissue in the reaction system was used as the enzyme activity unit U and the activity of complex IV was expressed as U·g^−1^ FW.

### 2.9. Determination of Mitochondrial Permeability Transition Pore (MPTP) and Matrix Ca^2+^ Content of F. solani

The opening degree of the mitochondrial permeability transition pore was determined by WU et al. [27]. Mitochondria were extracted as described in Section 2.8. The test solution consisted of 230 mmol·L^−1^ mannitol, 70 mmol·L^−1^ sucrose, and 3 mmol·L^−1^ HEPES (pH 7.4). A total of 2 mL of test solution was added to each extracted mitochondrial sample and the absorbance was measured at 540 nm after 2 min of reaction.

The Ca^2+^ content in the mitochondrial matrix was determined according to the method of Gazdag et al. [28]. Mitochondria were extracted as described in Section 2.8. A total of 2 mL of reaction solution (230 mmol·L^−1^ mannitol, 70 mmol·L^−1^ sucrose, 3 mmol·L^−1^ HEPES, pH7.4) was added to the mitochondria. After thorough mixing, arsenic azo III (AIII) was added to a final concentration of 50 μmol·L^−1^ as an indicator of calcium ions. The mitochondria were stained in the dark at 20 °C for 45 min and the absorbance was measured at 680 nm.

### 2.10. Determination of the Content of 5′-Triphosphate (ATP), 5′-Diphosphate (ADP), 5′-Monophosphate (AMP), and EC of F. solani Mycelia

To determine the ATP, ADP, and AMP contents, we used the method of Peng et al. [29]. First, 0.3 g fresh mycelium was added to 2 mL of 3% HClO_4_, ground in the ice bath, and centrifuged at 4 °C and 10,000 rpm for 10 min. Next, 1 mL supernatant was removed, adjusted to pH 3.0 with 20% KOH, then reacted at 4 °C for 30 min before being centrifuged at 10,000× *g* for 10 min at 4 °C and stored at−20 °C for testing.

All standards and samples were filtered through Teflon membranes with a pore size of φ 0.22 μm (Agilent, Santa Clara, CA, USA) and separated using an Agilent 1260 high-separation fast liquid chromatography system (2695, Waters, Milford, MA, USA) equipped with an Eclipse Plus C18 column (5 μm, 150 × 4.6 mm). The column temperature was 30 °C and the mobile phase was phosphate buffer solution (0.05 mol·L^−1^, pH 4.1) and methanol in buffer solution (99.9:0.1). The flow rate was 0.7 mL·min^−1^, the flow time was 15 min, and a UV/vis variable wavelength detector was used to measure the detection wavelength of 254 nm. Samples were 5 μL and were quantified by peak area using an external algorithm. The energy charge value was calculated using Formula (2):EC = (ATP + 1/2 ADP)/(ATP + ADP + AMP)(2)

### 2.11. Statistical Analysis

All of the determinations in this study were repeated at least three times. Data were expressed as means and standard errors, and Origin 18.0 (Northampton, MA, USA) was used for mapping. Significance and correlation analyses (*p* < 0.05) were performed using SPSS 26.0 (SPSS Inc., Chicago, IL, USA).

## 3. Results

### 3.1. Antimicrobial Effects of PGA

Colony diameter is a vital growth index reflecting pathogenic microorganisms. Our results showed that PGA significantly inhibited the colony growth of *F. solani*, with an average inhibition diameter of 10.48 ± 0.23 mm and an inhibition rate of 50.24 ± 0.12% (Figure 2A). There was no significant difference between methanol and sterile water control (results not shown).

Biomass is a crucial growth index to measure the growth and development of fungi. The results showed that the control *F. solani* grew well and had the characteristics of a typical microbial growth curve. The biomass increased with the prolongation of culture time. The biomass of the PGA treatment decreased first and then increased slightly with the prolongation of culture time (Figure 2B). Compared with the control, PGA treatment significantly decreased the biomass of *F. solani*. At 9 h and 36 h, they were respectively 69.53% and 65.54% lower than the control (*p* < 0.05).

### 3.2. Effect of PGA on Spore Germination of F. solani

Spore germination rate can reflect the strength of pathogen viability. The control germinated after 9 h, and the germination rate was 94.79%. The spore germination rate of *F. solani* treated with different concentrations of PGA was significantly different (Table 1) (*p* < 0.05). With the decrease in PGA concentration, the spore germination rate gradually increased. The inhibition rate of EC_50_ and 1/2 EC_50_ was the strongest, and the germination rate was unchanged with time. The spore germination rate of *F. solani* reached 83.17% after 12 h treatment with 1/16 EC_50_ PGA, showing a dose-dependent trend.

### 3.3. Effects of PGA Treatment on the Glucose and Pyruvate Contents of F. solani

Glucose is the substrate of the glycolytic pathway, and its content significantly affects pyruvate production in the metabolic pathway. During the culture period, the glucose content of *F. solani* in the control increased first and then decreased, while the glucose content in the PGA treatment decreased with the increase of culture time (Figure 3A). Compared with the control, the glucose content in *F. solani* decreased significantly during the culture period of PGA treatment, which was 77.5% and 45.0% lower than that of the control at 3 h and 12 h, respectively (*p* < 0.05).

Pyruvate is the end product of the glycolysis pathway and its content is affected by PFK and PK activity. Our results showed that the pyruvate content of *F. solani* in the control remained unchanged and stable during the culture period, while the pyruvate content of *F. solani* in the PGA treatment increased first and then decreased following a gentle trend (Figure 3B). During the incubation period, the pyruvic acid content of *F. solani* in the PGA treatment was always higher than the control, being 133.56% and 79.19% higher than the control at 3 h and 36 h, respectively (*p* < 0.05).

### 3.4. Effects of PGA Treatment on the HK, PK, and PFK Activities of F. solani

HK, PK, and PFK are all critical enzymes in the glycolysis pathway. Our results showed that the HK activity of the control increased first and then decreased, reaching the peak at 9 h. The HK activity of the PGA treatment decreased first, then increased, and then tended to be gentle (Figure 4A). During the incubation period, the HK activity of the PGA treatment was significantly lower than that of the control (*p* < 0.05), and was 82.14% and 77.75% lower than that of the control at 3 h and 12 h, respectively.

During the incubation period, the changing trend of PFK activity in the control and the PGA treatment was the same, increasing first and then decreasing (Figure 4B). The content of PFK in the PGA treatment was higher than that in the control at all time points except for 0, 3, and 24 h, and was 47.06% and 26.50% higher than in the control at 6 h and 12 h, respectively.

The PK activity of the control was unchanged during the culture process except for 24 h, where it was relatively gentle. The results of the PGA treatment were opposite to those of the control. Except for 24 h, the PK content was significantly higher than that in the control (Figure 4C) (*p* < 0.05). At 9 h and 36 h, the PK activity of the PGA treatment was 4.48 times and 4.28 times higher than that of the control, respectively.

### 3.5. Effect of PGA on the ATPase Activity of F. solani

Mitochondrial ATP content and EC are closely related to ATPase activity. Our results showed that the changes in Ca^2+^-ATPase activity in the control and the PGA treatment were the same, increasing first and then decreasing. During the culture period, the Ca^2+^-ATPase activity was significantly lower than that of the control except for at 3 h (Figure 5A) (*p* < 0.05). At 9 h and 36 h, the Ca^2+^-ATPase activity of the PGA treatment was 29.24% and 39.78% lower than that of the control, respectively. The changes in H^+^-ATPase activity in the control and the PGA treatment were the same, increasing first and then decreasing. The H^+^-ATPase activity of *F. solani* in the PGA treatment was significantly lower than that in the control during culture (Figure 5B). At 9 h and 12 h, the activity of the PGA treatment was 37.67% and 49.89% lower than that of the control, respectively.

### 3.6. Effect of PGA Treatment on the Complex Activity of F. solani

The mitochondrial respiratory chain is located in the mitochondrial inner membrane, which is composed of mitochondrial respiratory chain complex I (NADH oxidase), complex II (succinate-coenzyme Q reductase), complex III (CoQ-cytochrome C reductase), and complex IV (cytochrome C oxidase). The results showed that PGA treatment slightly affected the activity of *F. solani* complex I and II, although there was no significant difference (Figure 6A,B). PGA treatment significantly increased the activity of complex III by 28.08 times and 34.96% at 9 h and 36 h, respectively (Figure 6C) (*p* < 0.05). After *F. solani* was treated with PGA for 9 h, the activity of complex IV decreased. After 36 h, its activity was significantly lower than that of the control (Figure 6D) (*p* < 0.05).

### 3.7. Effects of PGA on Ca^2+^ Content and Permeability Transition Pore (MPTP) in the Mitochondrial Matrix of F. solani

The content of free calcium in the mitochondrial matrix mainly reflects the degree of apoptosis, while the change of OD_680_ reflects the level of Ca^2+^ concentration. Our results showed that the Ca^2+^ content in the control’s mitochondrial matrix remained steady, while the Ca^2+^ content in the PGA treatment showed an upward trend (Figure 7A). Compared with the control, the mitochondrial matrix Ca^2+^ content of the PGA treatment was significantly higher than that of the control after 9 h treatment, and was 26.14% and 48.06% higher than the control at 9 h and 12 h, respectively (*p* < 0.05). This indicates that PGA treatment resulted in excessive Ca^2+^ accumulation in the mitochondrial inner membrane, triggering apoptosis.

The opening degree of the MPTP can be used to measure the degree of damage to membrane permeability. The larger the absorbance value, the greater the opening degree of the mitochondrial permeability transition pore. The results showed that the MPTP openness of the control did not change significantly during the culture period, while the MPTP (OD_540_) openness with the PGA treatment gradually increased with the extension of the culture time (Figure 7B). The value during PGA treatment was significantly higher over the culture period than the control (*p* < 0.05). At 9 h and 36 h, the value for the PGA treatment was significantly higher than for the control by 98.26% and 154.37%, respectively (*p* < 0.05). This indicates that PGA treatment caused an increase in MPTP opening, ultimately leading to an increase in mitochondrial membrane permeability.

### 3.8. Effects of PGA on ATP, ADP, and AMP Content in F. solani

Decreased ATP content and energy charge levels mean that there is insufficient energy to maintain cell life activities, thereby accelerating apoptosis. Our results showed that the changing trend of ATP content in the control and the PGA treatment was the same, decreasing and then leveling off (Figure 8A). During the culture period, the ATP content of the PGA treatment was significantly lower than that of the control at 3 h and 12 h, at 73.26% and 82.43% lower than that of the control, respectively (*p* < 0.05).

During the incubation period, the content of ADP in the control was stable except for at 36 h, while it decreased and then leveled off in PGA treatment (Figure 8B). The ADP content in the PGA treatment was significantly lower than in the control (*p* < 0.05). At 3 h and 12 h, it was 64.50% and 70.19% lower than the control, respectively.

During the incubation period, AMP content in the control increased first and then decreased, while it decreased in the PGA treatment (Figure 8C). PGA treatment decreased the AMP content at 6 h and 12 h, where it was 59.32% and 75.49% lower than the control, respectively.

Compared with the control, the EC value was unstable and fluctuated obviously after PGA treatment (Figure 8D). The above results show that PGA treatment interfered with the energy metabolism of *F. solani*, resulting in intracellular ATP deficiency and metabolic disorders.

## 4. Discussion

In this experiment, the inhibitory effects of PGA on *F. solani* were comprehensively analyzed by assessing the changes in growth and development, glycolysis products and key enzymes, ATPase, mitochondrial complex, and mitochondrial structure. Colony diameter, biomass, and spore germination rate are critical indicators to measure the growth and development of pathogenic fungi. In this work, we found that PGA could significantly inhibit the colony growth, biomass, and spore germination rate of *F. solani*. *Streptomyces sioyaensis* treatment has been found to inhibit *Valsa sordida* colony growth, biomass, and spore germination, which is consistent with the results of this work [21].

The glycolysis pathway exists in organisms and can provide energy for life activities. Microorganisms transport glucose into cells through the glucose–phosphate transport system; it is converted into 6-phosphate glucose by phosphorylation and then produces pyruvate and ATP through the EMP pathway. It is one of microorganisms’ most important energy metabolism pathways and is the energy basis of bacterial life activities [30]. HK is the rate-limiting enzyme in the glycolytic pathway, and is the first key enzyme for the six-carbon sugar to enter the glycolytic pathway [31], while PFK is the second key enzyme in the glycolysis pathway [32]. PK is an important enzyme that can directly produce ATP. The higher its activity, the greater the demand for ATP [33]. In this work, we have shown that PGA treatment can reduce the glucose content and HK activity of *F. solani*, increase PFK and PK activity, and promote the accumulation of pyruvate. Previous studies have found that HK is an allosteric enzyme and that its allosteric subunits can be recognized by certain small molecule compounds in order to change its function, thereby regulating the EMP pathway [32,34]. N-acetyl-d-glucosamine was found to reduce peach resistance to *Monilinia fructicola* -induced brown rot by inhibiting HK activity, thereby affecting glucose metabolism [35]. This is consistent with the results of the present work. This may be due to HK being an allosteric enzyme, meaning that PGA can regulate its activity by binding to the allosteric subunit of HK. In this work, we found that PFK activity after PGA treatment was slightly higher than in the control, although the overall change was the same. PGA may not contain active substances that bind to PFK subunit allosteric sites. In this work, we have shown that PGA treatment can induce a rapid increase in the PK activity of *F. solani*. Pyruvate is the final product of glycolysis and can participate in different metabolic pathways; for example, in the electron transport chain process, pyruvate enters the tricarboxylic acid cycle under aerobic conditions and is completely oxidized to CO_2_ and H_2_O. Under anaerobic conditions, ethanol or lactic acid is produced [36]. In this study, we found that PGA treatment, increased pyruvate content, which corresponded to an increase in PK activity. Therefore, PGA may hinder pyruvate transport in tricarboxylic acid by inhibiting the tricarboxylic acid cycle and increasing its content. In summary, PGA treatment can reduce the glucose content of *F. solani* and promote the accumulation of pyruvate by affecting the activity of critical enzymes in glycolytic metabolism.

Energy is the basis of cell metabolic activity. Insufficient ATP content will damage the cell and mitochondrial membranes [14]. It has been found that the treatment of *Bacillus pumilus* HN-10 antimicrobial peptide P-1 reduced the ATP content of *Trichothecium roseum* mycelium and inhibited its growth [27]. O-vanillin treatment destroyed the mitochondrial membrane of *Aspergillus flavus*, resulting in increased ATP leakage and inhibited growth [37]. *Ferulago capillaris* essential oil has been found to significantly inhibit the activity of complex mitochondrial enzymes and energy production in yeast cells, thereby inhibiting growth [38]. In this work, PGA treatment significantly reduced the ATP content in *F. solani*, indicating that PGA harmed the permeability of the mitochondrial membrane, resulting in increased ATP leakage in mitochondria. ATPase regulates ion balance and energy metabolism inside and outside the mitochondrial membrane. H^+^-ATPase and Ca^2+^-ATPase are critical enzymes in cell respiratory metabolism [39]. H^+^-ATPase is an abundant protein on the cell membrane. It is a proton pump with multiple subunits. It both hydrolyzes ATP and acts as a proton pump. It can provide a driving force for the synthesis of ATP, and participates in many other physiological processes of pathogenic fungi, including stress response, mycelium growth, drug resistance, and more [40,41,42]. Mycelium development and reduced virulence of *Canidia Albicans* were found to be accompanied by loss of H^+^-ATPase activity [43]. Decreased H^+^-ATPase activity of *Penicillium digitatum* inhibited its pathogenicity to citrus fruits [44]. This is consistent with the results of the present work, in which the significant decrease of H^+^-ATPase activity by PGA treatment reduced the ability of *F. solani* to cope with stress. Ca^2+^-ATPase mainly catalyzes the hydrolysis of ATP inside the plasma membrane. This decrease in its activity inevitably affects the function of the Ca^2+^ pump, destroying the balance of intracellular Ca^2+^ and even leading to intracellular Ca^2+^ overload, resulting in apoptosis [45,46]. In this work, Ca^2+^-ATPase activity decreased significantly after PGA treatment of *F. solani*, which was consistent with the change in H+-ATPase activity. It has been reported that the decrease in the activity of the two leads to an insufficient cell energy supply, accelerating the decline of cells [44]. In addition, electron transfer on the mitochondrial respiratory chain mainly provides energy for oxidative phosphorylation, which is vital for the survival and maintenance of aerobic cells [47]. In this work, we found that the activity of mitochondrial complexes I, II, and III showed an upward trend after PGA treatment, among which the activity of complex III increased the most significantly, while the activity of complex IV decreased after PGA treatment. Based on the above results, it can be inferred that PGA reduces the activity of ATPase and complex IV of *F. solani*, causing insufficient ATP synthesis, breaking the ATP balance, and preventing the energy and critical substances required for cell growth and reproduction from being synthesized in a timely way, thereby achieving the effect of inhibiting growth.

Mitochondria are vital organelles that produce energy in eukaryotic cells. In addition, they regulate membrane potential, cell metabolism, and apoptosis signaling [48]. It has been reported that citral can exert antifungal activity by increasing the permeability of the mitochondrial membrane [14]. Ca^2+^ mainly exists in the inner membrane of mitochondria, which regulates mitochondrial metabolism and maintains the production of ATP to meet growth needs.

In this work, we found that PGA treatment resulted in the continuous inflow of Ca^2+^ into the mitochondrial matrix. This may be due to Ca^2+^-dependent phospholipid and enzyme-induced non-specific permeability changes in the mitochondrial inner membrane. It has been reported that changes in mitochondrial permeability are mainly due to the opening of non-specific channels in the mitochondrial inner membrane [49]. In this study, we found that PGA treatment led to the formation of non-specific channels in cells, resulting in increased openness of the MPTP. It has been reported that when these channels are formed the selective permeability of the membrane is destroyed, the inner membrane loses its barrier function, a large number of solute molecules enter the mitochondria, and the mitochondrial membrane potential decreases, eventually leading to disintegration of the mitochondrial structure [27].

## 5. Conclusions

PGA treatment inhibited the colony growth, biomass, and spore germination of *F. solani*. PGA treatment decreased glucose content and HK activity while increasing PFK and PK activity and promoting pyruvate accumulation. In addition, PGA inhibited the activity of mitochondrial complex IV and ATPase in *F. solani*. PGA treatment increased mitochondrial matrix Ca^2+^ content and increased the permeability transition pore (MPTP), causing mitochondrial membrane structure damage and dysfunction and reducing the ATP content and EC value of *F. solani*. In summary, the inhibitory effect of PGA on *F. solani* may be closely related to activation of the EMP pathway, destruction of mitochondrial structure, and a resulting reduction in energy charge level.

## Figures and Tables

**Figure 1 jof-09-00777-f001:**
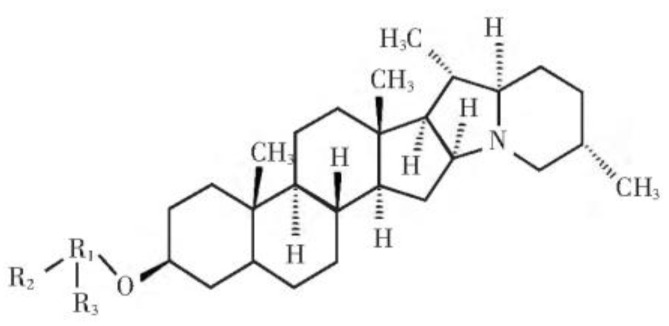
The chemical structure of PGA.

**Figure 2 jof-09-00777-f002:**
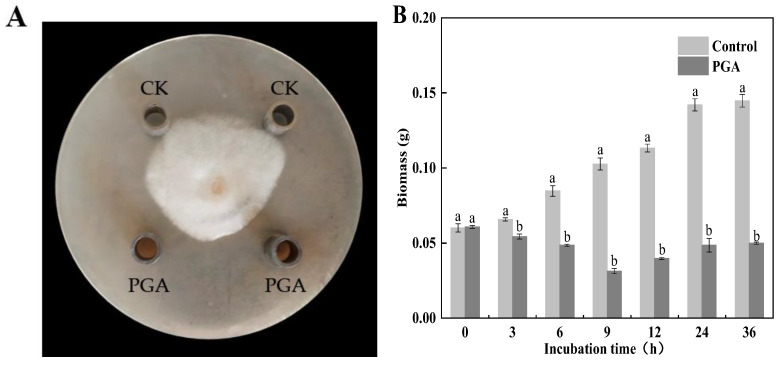
Effects of PGA treatment on the colony growth (**A**) and biomass (**B**) of *F. solani*. Note: PGA indicates the addition of PGA and CK indicates control. Different lowercase letters indicate significant differences between treatments at the same time (*p* < 0.05).

**Figure 3 jof-09-00777-f003:**
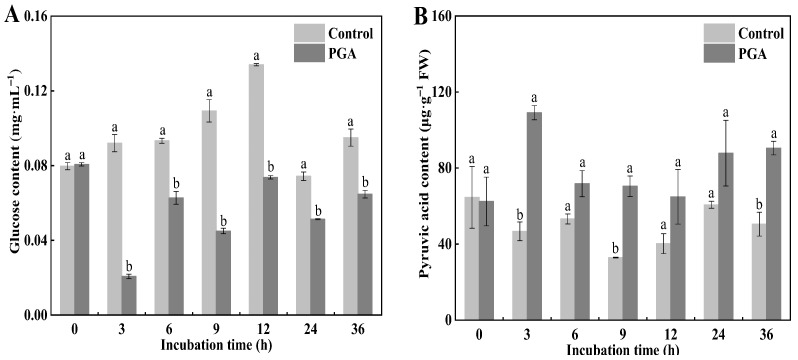
Effects of PGA treatment on the contents of glucose (**A**) and pyruvate (**B**) in *F. solani*. Different lowercase letters indicate significant differences between treatments at the same time (*p* < 0.05).

**Figure 4 jof-09-00777-f004:**
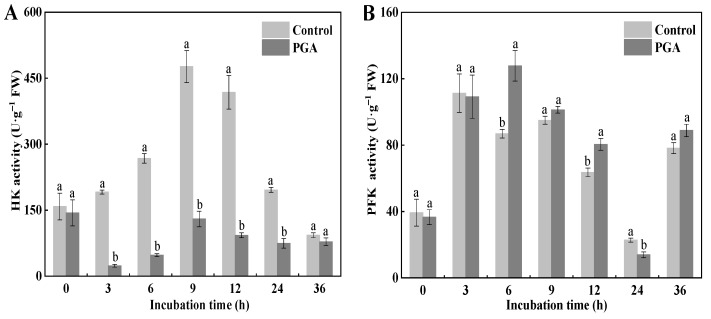
Effects of PGA on the activities of HK (**A**), PFK (**B**), and PK (**C**) in *F. solani*. Different lowercase letters indicate significant differences between treatments at the same time (*p* < 0.05).

**Figure 5 jof-09-00777-f005:**
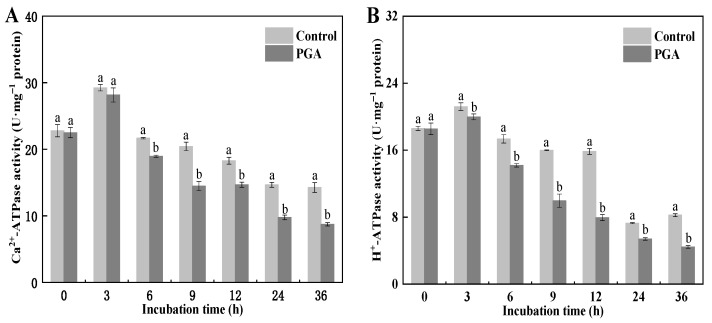
Effects of PGA on the activities of Ca^2+^-ATPase (**A**) and H^+^-ATPase (**B**) in *F. solani*. Different lowercase letters indicate significant differences between treatments at the same time (*p* < 0.05).

**Figure 6 jof-09-00777-f006:**
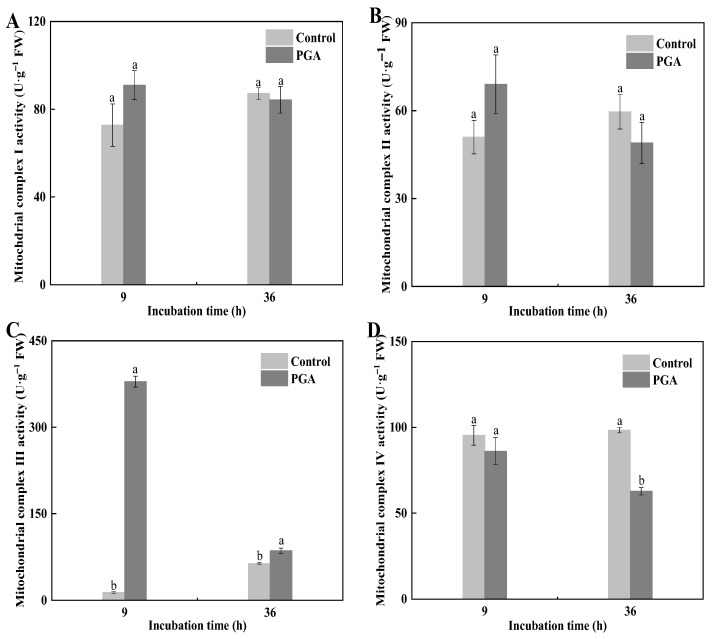
Effects of PGA treatment on activities of mitochondrial complex I (**A**), II (**B**), III (**C**), and IV (**D**) in *F. solani*. Different lowercase letters indicate significant differences between treatments at the same time (*p* < 0.05).

**Figure 7 jof-09-00777-f007:**
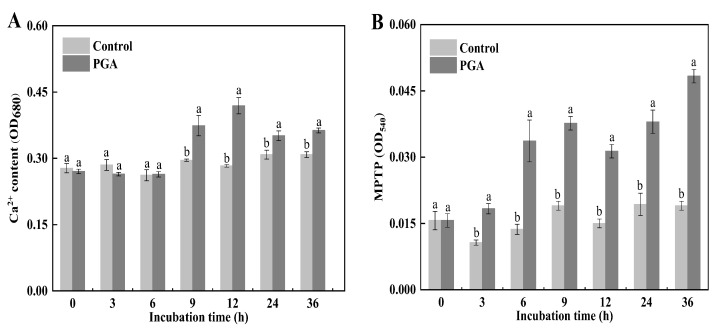
Effects of PGA treatment on mitochondrial Ca^2+^ content (**A**) and MPTP (**B**) of *F. solani.* Different lowercase letters indicate significant differences between treatments at the same time (*p* < 0.05).

**Figure 8 jof-09-00777-f008:**
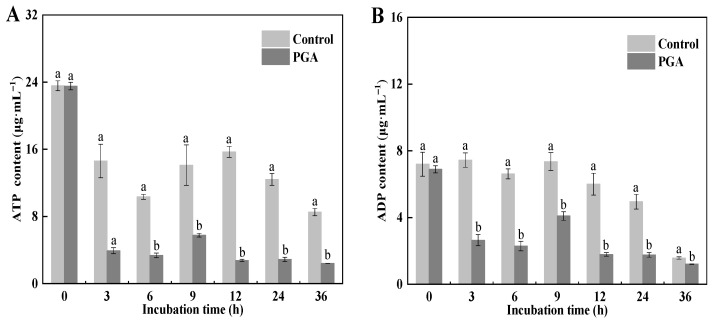
Effects of PGA on ATP (**A**), ADP (**B**), and AMP (**C**) contents and on EC (**D**) of *F. solani*. Different lowercase letters indicate significant differences between treatments at the same time (*p* < 0.05).

**Table 1 jof-09-00777-t001:** Effect of PGA treatment on spore germination of *F. solani*.

Treatment	Rate of Spore Germination (%)				
	0 h	3 h	6 h	9 h	12 h
Control	2.84 ± 0.38 a	3.80 ± 1.69 a	70.96 ± 1.82 a	94.79 ± 2.03 a	97.95 ± 0.58 a
EC_50_	2.94 ± 0.34 a	1.71 ± 0.42 b	3.40 ± 0.42 d	3.14 ± 0.46 e	2.70 ± 0.57 e
1/2 EC_50_	2.71 ± 0.59 a	1.86 ± 0.74 b	5.10 ± 0.15 d	3.01 ± 1.17 e	3.55 ± 1.01 e
1/4 EC_50_	2.32 ± 0.89 a	2.13 ± 0.40 ab	5.02 ± 0.84 d	22.62 ± 5.65 d	44.04 ± 3.67 d
1/8 EC_50_	2.68 ± 0.33 a	2.27 ± 0.40 ab	13.31 ± 1.82 c	36.60 ± 2.28 c	60.24 ± 0.82 c
1/16 EC_50_	2.68 ± 0.39 a	2.76 ± 0.48 a	19.28 ± 2.56 b	54.78 ± 4.88 b	83.17 ± 1.71 b

Different lowercase letters indicate significant differences between treatments at the same time (*p* < 0.05).

## Data Availability

Not applicable.

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
