# Peer review of "Effect of Potato Glycoside Alkaloids on Energy Metabolism of Fusarium solani"

_jof, 2023, doi:10.3390/jof9070777_

Round 1
Reviewer 1 Report
Article: “Effect of potato glycoside alkaloids on energy metabolism Fusarium solani”
1) “Fusarium solani was isolated from the diseased plants of root rot of wolfberry”. Add the author of the scientific name of the fungus. Add fungus family name. How identified the taxonomy of the specimen. 2) “The green potato peels and buds were dried in a vacuum blast..”. Give the scientific name of the plant species potato. Where it is cultivated. 3) “The mycelium plug(5mm) was placed in the center of potato dextrose agar (PDA)”. Mycelium of what?
4) “…..from the mycelium plug 20 μL PGA”. How can PGA form mycelium? 5) “Fusarium solani is one of the primary pathogens causing root rot of wolfberry. The aims of this study were to investigate the inhibitory effect of potato glycoside alkaloids (PGA) on F. solani and its energy metabolism”. The inhibitory effect of PGA and the mechanism of the action on F. solani was well known through some previous studies, such as below; what is the new or novelty in this work.
He, J.; Duo T.T.; Chen, W.; Zhang, X.Y. Mechanism of Action of Potato Glycoalkaloids against Fusarium solani. Int J Agric Biol. 4932021, 25, 873‒880. DOI: 10.17957/IJAB/15.1741
Olsson, K., 1987. The influence of glycoalkaloids and impact damage on resistance to Fusarium solani var. coeruleum and Phoma exigua var. foveata in potato tubers. Journal of phytopathology, 118(4), pp.347-357.
Minor editing of English language required
Author Response
Major:
Point 1: “Fusarium solani was isolated from the diseased plants of root rot of wolfberry”. Add the author of the scientific name of the fungus. Add fungus family name. How identified the taxonomy of the specimen.
Response 1: We have added the more information of the fungus in revised manuscript as below.
Fusarium solani (Mart.) App.et Wollenwwas belongs to Tuberculariaceae. F. solani was identified by morphology and EF-1α gene sequence analysis and comparative identification.
Point 2: “The green potato peels and buds were dried in a vacuum blast.”. Give the scientific name of the plant species potato. Where it is cultivated.
Response 2: We have added the Latin name and varieties of potatoes, harvest place as your suggestion.
The potato (Solanum tuberosum L. cv. Longshu No. 9) were harvested from Tongwei County of Dingxi City, Gansu Province in China in October 2022. Fresh tubers were washed and exposed to sunlight for several weeks to accelerate the process of greening and germination. The green peels and buds were dried in a vacuum blast drying oven (80 °C, 12 h), ground using a grinder, and passed through an 80-mesh sieve. The obtained experimental samples were extracted by the acetic acid-ammonia precipitation method. PGA was dissolved in methanol and stored at 4 °C for later use.
Point 3: “The mycelium plug(5mm) was placed in the center of potato dextrose agar (PDA)”. Mycelium of what?
Response 3: The mycelium plug is F. solani. We have changed “The mycelium plug (5mm) was placed in the center of potato dextrose agar (PDA)” to “The mycelium plug (5mm) of F. solani was placed in the center of potato dextrose agar (PDA)”.
Point 4: “…..from the mycelium plug 20 μL PGA”. How can PGA form mycelium?
Response 4: The mycelium plug(5mm) of F. solani was placed in the center of potato dextrose agar (PDA), and four Oxford cups were placed evenly 2 cm away from the mycelium plug. The PGA was added dropwise (20 μL 88.10mg·mL-1) to each Oxford cup, and the same amount of sterile distilled water was used as a control and incubated in a 25 °C incubator for 7 days.
Point 5: “Fusarium solani is one of the primary pathogens causing root rot of wolfberry. The aims of this study were to investigate the inhibitory effect of potato glycoside alkaloids (PGA) on F. solani and its energy metabolism”. The inhibitory effect of PGA and the mechanism of the action on F. solani was well known through some previous studies, such as below; what is the new or novelty in this work.
He, J.; Duo T.T.; Chen, W.; Zhang, X.Y. Mechanism of Action of Potato Glycoalkaloids against Fusarium solani. Int J Agric Biol. 4932021, 25, 873‒880. DOI: 10.17957/IJAB/15.1741
Olsson, K., 1987. The influence of glycoalkaloids and impact damage on resistance to Fusarium solani var. coeruleum and Phoma exigua var. foveata in potato tubers. Journal of phytopathology, 118(4), pp.347-357.
Response 5: The reference of He et al. (2020) is the previous research results of our team. It mainly studied the antifungal mechanism of PGA against F. solani from the perspective of mycelium morphology and substance metabolism. Olsson et al. (1987) only studied the effect of PGA on potato rot during storage, but did not deeply study its antifungal mechanism. In this study, we studied the effect of PGA on the mitochondrial structure and energy level of F. solani from the perspective of energy metabolism to elucidate its antifungal mechanism.
We have tried our best to improve the manuscript and have made some changes in the manuscript. We marked with red where we made the changes. These changes do not affect the content and scope of the manuscript. We hope that the correction we have made will meet this requirement.
Thank you for your consideration. I look forward to hearing from you soon.
With best regards,
Prof. Jing He
College of Forestry,
Gansu Agricultural University,
Lanzhou 730030, China.
Email: hejing268@aliyun.com

Reviewer 2 Report
Authors must to provide more information about the chemical characterization of PGAs, and also the applied protocol yo obtain the respective extracts. For the introduction it Is recommended to include a schematic figure about the chemical structures of PGAs.
The methodology employed chemical procedures to detect, direct or indirectly glucose or usually reductor sugars. However, potatoes could contain non-reductor sugar with an alkaloid moiety, how the authors discard this kind of sugars?
I suggest to explore molecular dynamics to explain the enzymatic inhibition results. This additional calculations could improve the quality of the manuscript.
Author Response
Major:
Point 1: Authors must to provide more information about the chemical characterization of PGAs, and also the applied protocol yo obtain the respective extracts. For the introduction it Is recommended to include a schematic figure about the chemical structures of PGAs.
Response 1: We have added the more information of PGA in troduction as below.
PGA is the ligand of solanidine with glucose, lactose and rhamnose, respectively. The crystal is white light-emitting needle-like, easily soluble in methanol, hot ethanol, slightly soluble in cold ethanol, almost insoluble in water, ether and benzene, melting point 248 °C, about 280-285 °C decomposition, and it can be hydrolyzed by acid and decomposed into solanidine and sugar. The structure is shown in Figure 1.
Figure 1
PGA were extracted by the acetic acid-ammonia precipitation method. 100 g of the potato sample was mixed with 400 mL of 5% acetic acid, stirred for 60 min, and filtered. The residue was extracted twice with 200 mL of 5% acetic acid, and the filtrate was combined, and its pH adjusted to 11 with ammonia. After extracting three times with 200 mL of water-saturated n-butanol, the extracts were combined and dried on a rotary evaporator, and the residue mixed with 20 mL of methanol to obtain total glycoalkaloids extract.
Point 2: The methodology employed chemical procedures to detect, direct or indirectly glucose or usually reductor sugars. However, potatoes could contain non-reductor sugar with an alkaloid moiety, how the authors discard this kind of sugars?
Response 2: During the extraction of PGA, PGA was dissolved in water-saturated n-butanol solution. The surface of the sugar structure is almost all hydrophilic hydroxyl groups, while one end of the n-butanol molecule has a hydroxyl group and the other end is a hydrophobic alkyl group. Therefore, the solubility of sugar in water-saturated n-butanol is not as good as that in water during the extraction process. Sugars are dissolved in waste aqueous solutions.
The extracted sugars are dissolved into the lower solvent, so it can be excluded discard this kind of sugars. Therefore, PGA does not contain sugar. During the experiment, we washed the treated mycelium several times, and then measured the glucose content of F. solani mycelium. Therefore, it has no effect on the experimental results.
Point 3: I suggest to explore molecular dynamics to explain the enzymatic inhibition results. This additional calculations could improve the quality of the manuscript.
Response 3: Thanks for your suggestion. Molecular dynamics can be used to accurately explain enzyme activity. However, at present, the kit is also widely used in the determination of enzyme activity due to its advantages of fast, operation and accuracy. For example, the following articles. We will use molecular dynamics methods to determine enzyme activity according to your suggestion in subsequent experiments.
Wu, S.J.; Yun, J.M.; Wang, R.; Zhang, W.W.; Hao, L.; Pei, P.Z. Analysis of the effects of antifungal peptide P-1 from Bacillus pumilus HN-10 on energy metabolism of Trichothecium roseum. Food Biosci. 2022, 47, 101668. DOI: 10.1016/j.fbio.2022.101668
Zheng, S.; Jing, G.; Wang, X.; Ouyang, Q.; Jia L.; Tao, N. Citral exerts its antifungal activity against Penicillium digitatum by affecting the mitochondrial morphology and function. Food Chem. 2015, 178, 76-81. DOI: 10.1016/j.foodchem.2015.01.077
Li, Q.; Zhao, Y.; Zhu, X.; Xie, Y. Antifungal effect of o-vanillin on mitochondria of Aspergillus flavus: ultrastructure and TCA cycle are destroyed. Int. J. Food Sci. Technol. 2022, 57, 3142-3149. DOI: 10.1111/ijfs.15647
We have tried our best to improve the manuscript and have made some changes in the manuscript. We marked with red where we made the changes. These changes do not affect the content and scope of the manuscript. We hope that the correction we have made will meet this requirement.
Thank you for your consideration. I look forward to hearing from you soon.
With best regards,
Prof. Jing He
College of Forestry,
Gansu Agricultural University,
Lanzhou 730030, China.
Email: hejing268@aliyun.com

Round 2
Reviewer 1 Report
1. "The fungus is the main pathogen of plant soil-borne diseases and could cause a variety of plant rot, such as root rot, stem rot, ear rot, and stem base rot". Ear rot of which plant?
2. "The green potato peels and buds were dried in a vacuum blast drying oven (80 °C, 12 h), ground using a grinder, and passed through an 80-mesh sieve". How do you identify that this harvested potato contains glycoside alkaloids?
3. “After extracting three times with 200 mL of water-saturated n-butanol, the extracts were combined and dried on a rotary evaporator, and the residue mixed with 20 mL of methanol to obtain total glycoalkaloids extract”. Which analytical method was used to identify glycoalkaloids?
Minor editing of English language required
Author Response
Major:
Point 1: "The fungus is the main pathogen of plant soil-borne diseases and could cause a variety of plant rot, such as root rot, stem rot, ear rot, and stem base rot". Ear rot of which plant?
Response 1: It has been reported that Fusarium solani can cause ear rot of maize (Morales-Rodríguez et al., 2007).
Morales-Rodríguez, I.; Yanez-Morales, M.J.; Silva-Rojas, H.V., García-de-los-Santos, G.; Guzman-de-Pena, D.A. Biodiversity of Fusarium species in mexico associated with ear rot in maize, and their identification using a phylogenetic approach. Mycopathologia, 2007, 163, 31-39. DOI 10.1007/s11046-006-0082-1
Point 2: "The green potato peels and buds were dried in a vacuum blast drying oven (80 °C, 12 h), ground using a grinder, and passed through an 80-mesh sieve". How do you identify that this harvested potato contains glycoside alkaloids?
Response 2: The research shows that potatoes are easy to cause green skins and promote germination after light treatment, and potato glycoside alkaloids mainly exist in green skins and buds (Percival et al., 1996; Benkeblia, 2019).
Percival, G.; Dixon, G.R.; Sword, A. Glycoalkaloid Concentration of Potato Tubers Following Exposure to Daylight. J. Sci. Food Agric. 1996, 71, 69-63. DOI: 10.1002/(sici)1097-0010(199605)7
1:1<59: aid-jsfa548>3.0.co;2-1
Benkeblia, N. Potato Glycoalkaloids: occurrence, biological activities and extraction for biovalorisation – a review. Int. J. Food Sci. Technol. 2019, 55, 2305-2313, DOI:10.1111/ijfs.14330
Point 3: “After extracting three times with 200 mL of water-saturated n-butanol, the extracts were combined and dried on a rotary evaporator, and the residue mixed with 20 mL of methanol to obtain total glycoalkaloids extract”. Which analytical method was used to identify glycoalkaloids?
Response 3: PGA was identified by high performance Liquid chromatography (HPLC). 0.0625、0.125、0.25、0.5、1、2mg·mL-1 standard solution of α-solanine and α-chaconine was prepared. The results were obtained by measuring at 202 nm with a UV-detector and by comparing the regions covered by the standards of α-solanine and α-chaconine(Kasnak et al., 2018).
Kasnak, C.; Artik, N. Change in Some Glycoalkaloids of Potato under Different Storage Regimes. Potato Res. 2018, 61, 183-193. DOI: 10.1007/s11540-018-9367-2
We have tried our best to improve the manuscript and have made some changes in the manuscript. We marked with red where we made the changes. These changes do not affect the content and scope of the manuscript. We hope that the correction we have made will meet this requirement.
Thank you for your consideration. I look forward to hearing from you soon.
With best regards,
Prof. Jing He
College of Forestry,
Gansu Agricultural University,
Lanzhou 730030, China.
Email: hejing268@aliyun.com

Reviewer 2 Report
Authors performed modifications on the manuscript, achieving a better version. I suggest to publish this manuscript in Jornal of fungi
Author Response
Dear reviewer:
Thank you very much for your recognition of this study.
With best regards,
Prof. Jing He
College of Forestry,
Gansu Agricultural University,
Lanzhou 730030, China.
Email: hejing268@aliyun.com